# LLM+A: Grounding Large Language Models in Physical World with Affordance Prompting

## Abstract

While large language models (LLMs) are successful in completing various language processing tasks, they easily fail to interact with the physical world properly such as generating control sequences. We find that the main reason is that LLMs are not grounded in the physical world. Existing LLM-based approaches circumvent this problem by relying on additional pre-defined skills or pre-trained sub-policies, making it hard to adapt to new tasks. In contrast, we aim to address this problem and explore the possibility to prompt pre-trained LLMs to accomplish a series of robotic manipulation tasks in a training-free paradigm. Accordingly, we propose a framework called LLM+A(ffordance), where the LLM serves as both the sub-task planner (that generates high-level plans) and the motion controller (that generates low-level control sequences). To ground these plans and control sequences on the physical world, we develop the *affordance prompting* technique that stimulates the LLM to 1) predict the consequences of generated plans and 2) generate affordance values for relevant objects. Empirically, we evaluate the effectiveness of LLM+A in various robotic manipulation tasks with natural language instructions and demonstrate that our approach substantially improves performance by enhancing the feasibility of generated plans and control.

## 1 Introduction

Recent large language models (LLMs) (Ouyang et al., 2022; Chowdhery et al., 2022; Brown et al., 2020; Chung et al., 2022) have exhibited remarkable capabilities in a wide range of natural language processing (NLP) tasks such as daily dialogue (Thoppilan et al., 2022; Adiwardana et al., 2020), programming (Chen et al., 2021; Li et al., 2023), and text writing (Touvron et al., 2023). As these pre-trained models assimilate extensive knowledge from internet-scale text corpora and various domain-specific datasets, they can be leveraged as foundation models and provide rich prior knowledge. Owing to these advancements, new paradigms have emerged, employing LLMs as artificial intelligence (AI) assistants to perform embodied robotic tasks in real-world settings (Driess et al., 2023; Brohan et al., 2023; Vemprala et al., 2023).

We study the problem of language-conditioned robotic manipulation control with LLMs. Existing approaches can be generally classified into two main categories: employing LLMs as high-level sub-task planners (Ahn et al., 2022; Huang et al., 2022a; Wang et al., 2023b), or directly training an end-to-end large model as low-level motion controllers (Driess et al., 2023; Brohan et al., 2023; Mu et al., 2023). The first category leverages the powerful planning and reasoning capabilities of LLMs, enabling them to decompose human instructions into a series of textual sub-tasks. Nevertheless, to execute these decomposed plans in real-world scenarios, current methods still depend on pre-trained skills or primitive actions (Ahn et al., 2022; Huang et al., 2023b; 2022b; Singh et al., 2023), which are usually learned by behavior cloning and reinforcement learning techniques. This reliance on specific sub-policies can limit the applicability of these approaches, as they necessitate vast amounts of robotic data and often struggle to generalize to unseen environments (Huang et al., 2023b) and different embodiments (Bousmalis et al., 2023). The second category typically trains a large-scale multi-task backbone model that integrates both linguistic and visual modalities to generate end-to-end control sequences (Brohan et al., 2023; Driess et al., 2023; Reed et al., 2022). However, the development of such models requires extensive multi-modal datasets, encompassing a diverse array of robotic tasks. This is costly for many researchers. Therefore, considering the impressive commonsense knowledge and powerful in-context learning abilities demonstrated by LLMs, can LLMs

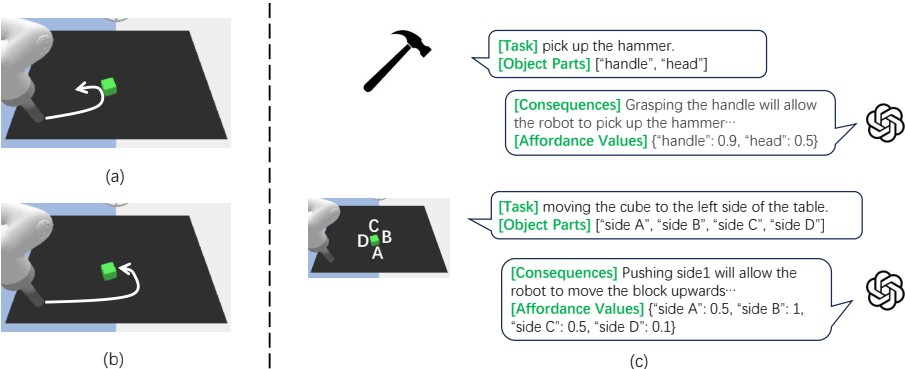

Figure 1: Consider the task of "Push the cube to the left side of the table". When the control sequences generated from LLMs are not grounded in the physical world, the robot will move to the left side of the cube to push it to the left (a) instead of the right location (b). This is due to the gap between the physical world and generated language plans. This gap can be bridged by prompting LLMs to predict execution consequences and goal-conditioned affordance values (c) in the proposed LLM+A method.

function as both sub-task planners and motion controllers, thereby addressing robotic manipulation tasks without the need for additional training?

However, how to exploit commonsense from LLMs to real-time fine-grained robotic manipulation tasks remains a challenging problem. This difficulty primarily comes from the fact that LLMs are not grounded in the physical world, which can potentially result in erroneous or inexecutable plans (Ahn et al., 2022; Wang et al., 2023a; Yao et al., 2022). For instance, consider a tabletop robotic arm situated to the left of a block, and the instruction is "push the block to the left side of the table". It may move directly right and then push the block as shown in Figure 1(a) instead of maneuvering around to the right side of the object as shown in Figure 1(b). In this scenario, the block will not move as expected. The main reason is that LLMs lack more comprehensive information regarding the current environment, such as spatial relationships on objects. Besides, pre-trained LLMs neglect to comprehend the consequences of the generated plan in the actual physical world. Therefore, to generate executable control sequences, LLMs must consider physical laws and object functionalities, which can help outline the possibilities and effects of generated actions for robots interacting with objects or specific environmental features.

In the field of robotics, the concept of *affordance* is considered a crucial mechanism to enable robots to understand and interact with environments. Given the task instruction, affordance values indicate the functional priorities for the robot of objects in the current environment to complete the task. At the current stage of affordance research, the related prior knowledge is usually provided by humans (Yang et al., 2023). In contrast to previous studies, we demonstrate that LLMs are proficient at predicting execution consequences and inferring affordance values of different object parts as shown in Figure 1(c), which effectively provides robots with useful information to complete instructions. For instance, within the context of hammering or pushing tasks, LLMs can interpret human directives and ascertain task-specific executable parts for grasping, functioning, effecting, or pushing respectively.

Motivated by this concept, we introduce a framework, called LLM+A, to exploit the extraction of embodied commonsense and reasoning capabilities of LLMs to generate control sequences for robotic manipulation tasks following textual instructions in a training-free pradigm. Firstly, we employ vision language models (VLMs), such as open-vocabulary detectors (Gu et al., 2021; Minderer et al., 2022), or large multimodal models (Gao et al., 2023; Awadalla et al., 2023), to provide textual observation of target objects (such as object shapes, colors, and positional relationships) and interactive environments to the LLMs. Then, we obtain goal-conditioned affordance values from LLMs that describe the priorities of object executable parts for interaction via *affordance prompting*. Based on the above visual perception and affordance values, LLMs decompose human instructions into high-level sub-tasks, which are feasible in the physical world. Subsequently, LLMs also generate control

sequences for the current task. Our experiments demonstrate that grounding LLMs in the physical world to generate motion plans via unlocking their affordance knowledge can highly enhance the performance in robotic manipulation tasks compared to non-grounded baselines.

Our contributions are summarized as follows:

- We propose LLM+A that adopt large language models (LLMs) to serve as both the high-level sub-task planner and the low-level motion controller in robotic control tasks in a training-free paradigm.

- To improve the physical executability of both the sub-task plans and the control sequences generated while adhering to the language instruction, we propose *affordance prompting* to stimulate the ability of LLMs to infer goal-conditioned affordance values, which indicate the executable priorities of different parts of interacted objects.

- Experimental results on heterogeneous robotic tasks validate the effectiveness and robustness of our method.

## 2 RELATED WORK

**LLMs for Sub-Task Planning and Motion Controlling.** With the development of transformers in recent years, pre-trained large language models (Ouyang et al., 2022; Chowdhery et al., 2022; Brown et al., 2020; Chung et al., 2022) have become an active area of research. These models, pre-trained on vast amounts of internet-scale text corpora from various tasks, exhibit remarkable commonsense and reasoning capabilities in embodied tasks. Numerous recent approaches successfully employ LLMs to decompose abstract and human instructions into natural language-based high-level plans (Ahn et al., 2022; Huang et al., 2022a;b; Ding et al., 2023; Shah et al., 2023; Min et al., 2021) or code-based plans (Liang et al., 2023; Huang et al., 2023a). For instance, ZSP (Huang et al., 2022a) demonstrate that LLMs can be utilized for task planning in household domains through iteratively augmented prompts, enabling the semantic translation of plans into admissible skills. Similarly, Say-Can (Ahn et al., 2022) leverage LLMs to facilitate robot task planning by incorporating affordance functions to ensure plan feasibility. While these methods show surprising zero-shot generalization ability of task planning, the execution of decomposed plans remains dependent on pre-trained skills, which are usually acquired via behavior cloning and reinforcement learning. This reliance may limit their applicability, as the training process necessitates substantial amounts of robotic data, which is costly to obtain. On the other hand, some general robotic models have been proposed to achieve end-to-end control for real-world robotic applications (Brohan et al., 2023; 2022; Driess et al., 2023; Mu et al., 2023; Stone et al., 2023; Jang et al., 2022; Suglia et al., 2021). These methods benefit from high-capacity networks and open-ended task-agnostic training with diverse datasets. In contrast, the proposed LLM+A leverages LLMs as both the sub-task planner and the motion controller in a training-free paradigm in robotic manipulation tasks.

**Affordance for Robotics.** As a popular concept proposed in the field of psychology, affordance has been extensively utilized in robotic tasks to facilitate agents' comprehension and interaction with dynamic environments (Jamone et al., 2016; Xu et al., 2021a; Wu et al., 2021; Mo et al., 2021). Briefly, affordance encapsulates the potential outcomes and effects resulting from robot's actions on a specific object or, more broadly, a segment of the environment. Existing research can be divided into three primary categories: modeling action possibilities (Mo et al., 2021; Borja-Diaz et al., 2022; Yang et al., 2021), generating keypoint affordances (Fang et al., 2020; Manuelli et al., 2019; Qin et al., 2020; Xu et al., 2021b), and learning partial dynamic models exclusively for afforded actions (Khetarpal et al., 2020; Xu et al., 2021a; Khetarpal et al., 2021). Since their development primarily relies on data-driven approaches within the visual domain, these methods exhibit limitations when applied to language-conditioned scenarios. Recent methods begin to leverage LLMs to obtain language-conditioned affordance values. For example, HULC++ (Mees et al., 2023) employed a self-supervised visual-lingual affordance model to guide robots toward actionable areas referenced by language. VoxPoser (Huang et al., 2023b) extracted affordances and constraints for robotic manipulation tasks from pre-trained LLMs, demonstrating generalizability to open-set instructions. In this work, we concentrate on grounding LLMs in more fine-grained robotic tasks by predicting execution consequences and extracting physical affordance values.

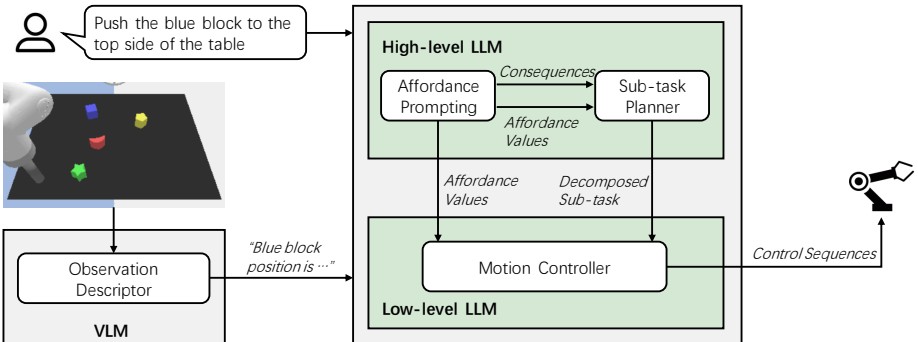

Figure 2: Overview of LLM+A. Given language instructions and image observations, LLM+A produces sub-task plans and control sequences for robotic control tasks. LLM+A is composed of a VLM and a hierarchical LLM. The VLM serves as an observation descriptor to provide the environment perception to the LLM. The high-level LLM is responsible for sub-task planning and the low-level LLM for motion controlling. Notably, the affordance values from the high-level LLM are necessary intermediate information for the LLM to understand the effects of potential actions and generate feasible plans grounded in the physical world.

## 3 METHOD

In this section, we first introduce the formulation of robotic manipulation tasks (Sec. 3.1). Then, we describe our LLM+A framework where the pre-trained visual-language model (VLM) serves as the observation descriptor and the large language model (LLM) serves as the high-level sub-task planner and the low-level motion controller (Sec. 3.2). Later, we introduce *affordance prompting* to predict consequences and generate affordance values to bridge the gap between generated plans/control sequences and the physical world (Sec. 3.3).

### 3.1 ROBOTIC MANIPULATION

In robotic manipulation tasks that we consider in this paper, the LLM-based agent needs to generate control sequences for a tabletop robotic arm to complete a given task instructed in open-vocabulary natural language based on the image observation. Specifically, on the $t$-th time step, the agent $\pi(a_t|o_t, l)$ perceives the image observation $o_t$ and outputs the action $a_t$ to follow the instruction $l$. The instruction $l$ is not constrained by any templates, grammatical structures, or vocabularies. For example, the instruction can be "push the red block to the left center side of the table" or "separate the yellow block and the green block". In this paper, the action of the LLM-based agent is a planned path segment of the end-effector of the robotic arm, represented by $K$ coordinates specifying the waypoints of the end-effector, i.e., when $a_t = ((x_1, y_1), \cdots, (x_K, y_K))$, the end-effector will move along the path $(x_1, y_1) \rightarrow \cdots \rightarrow (x_K, y_K)$.

### 3.2 LLM+A

In the LLM+A framework, we aim to leverage the commonsense knowledge and reasoning/planning capability from both the pre-trained VLM and LLM to complete robotic manipulation tasks. Further, we develop *affordance prompting* to incorporate the concept of affordance into zero-shot prompting for the LLM, which can be regarded as an extension of chain-of-thought (CoT) to embodied robotics. We present the LLM+A framework in Figure 2 and introduce different modules of LLM+A as follows.

**Observation Descriptor.** In this module, we feed the current image observation $o_t$ combined as a designed prompt template to the VLM-based observation descriptor to generate text description $s_t$ to provide necessary information needed in the subsequent decision making process of the LLM-

based sub-task planner and motion controller. Specifically, the text description $s_t$ contains the spatial location of relevant objects and all functional parts of them such as the four sides of the cube.

**Sub-task Planner.** In this module, we feed the text description of the current observation $s_t$ as well as the instruction $l$ to the sub-task planner to obtain a series of planned high-level sub-tasks $g = (g_1, g_2, \cdots)$ required to accomplish the task based on the current observation. Notice that this plan can change on the subsequent time steps based on future observations and the agent typically only executes the first sub-task on the current step. Further, to incentivize the sub-task planner to generate feasible plans, we develop the *affordance prompting* technique that instructs the sub-task planner to predict the expected consequences of the control and generate affordance values for the functional parts of relevant objects. We present the simplified version of the prompt template as follows:

> Template for Sub-Task Planner:
> *You are a robotic arm on the table which can [arm skills].*
> *You need to accomplish a series of robotic manipulation tasks: [guidelines].*
> *The task instruction is [task instruction].*
> *The objects on the table are [object parts].*
> *You need to:*
> *1. output the **consequences** of potential actions;*
> *2. output the **affordance values** of each object parts considering the potential consequences;*
> *3. output the **decomposed sub-tasks** according to the consequences and affordance values.*

In the template, *[arm skills]* describe the functions of the type of the arm; *[guidelines]* describe guidelines or contexts of the task such as the orientation of the table, the range/orientation of the spatial coordinates; *[task instruction]* is the open-vocabulary natural-language-based instruction $l$; *[object parts]* are text description of the observation $s_t$ generated by the observation descriptor. At last, we ask the LLM to decompose the task instruction $l$ into sub-tasks $g$. *Decomposed sub-tasks* refer to more specific plans appropriate for the robot to execute starting from the current state, e.g., "approach right side of the green block while avoiding the red block" and "push the green block to the top". To generate more feasible sub-tasks, we adopt *affordance prompting* which is highlighted in red. *Consequences* refer to the effects of object parts after the physical interaction by possible skills of the robot and *affordance values* indicate the extent to which each part of the object is expected to be interacted. These concepts will be further explained later in Sec. 3.3.

**Motion Controller.** In this module, given the decomposed sub-tasks $g = (g_1, g_2, \cdots)$ and the affordance values associated with different object parts, we ask the LLM to generate the specific action $a_t$. We also present the simplified version of the prompt template of this procedure as follows:

> Template for Montion Controller:
> *You are a robotic arm on the tabletop which can [arm skills].*
> *You need to accomplish a series of robotic manipulation tasks: [guidelines].*
> *The task instruction is [task instruction].*
> *The objects on the table are [object parts].*
> *Given the [decomposed sub-tasks] and the [affordance values], you need to output the **control sequence**.*
> *Note that [notes].*
> *The examples are as follows: [examples].*

In this template, *[decompoased sub-tasks]* and *[affordance values]* are the outputs generated by the sub-task planner. The control sequence is the action of the LLM-based agent which refers to a series of the waypoint coordinates of the end-effector in our case. In addition, we also provide *[notes]* and *[examples]* to facilitate the LLM to generate better control sequences. *[Notes]* can include formatting instructions such as "You need to generate the above outputs with JSON format", and *[examples]* follows the few-shot prompting practice.

### 3.3 Affordance Prompting

In robotic tasks, the concept of affordance plays a crucial role in enabling robots to comprehend and interact with the corresponding physical environment. This generally depends on prior knowledge of relevant actions and the task instruction. LLMs are highly proficient at inferring affordance values, owing to their rich commonsense knowledge learned from diverse robotic-related datasets during pre-training. For example, as shown in Figure 1(c), LLMs can accurately assess different parts of the hammer and their affordance values for the grasping task, based on which robot motions can be effectively suggested. Another example is a robotic manipulation task, where LLMs identify the most actionable edge with the highest affordance value for pushing the cube to the left side of the table. Motivated by this observation, the goal of our *affordance prompting* technique is to unlock LLMs with the ability to generate goal-conditioned affordance value, serving as a feasible intermediate reasoning step that assists the robot in understanding action priorities.

In LLM+A, given *[arm skills]*, *[task instruction]*, and *[object parts]*, we firstly query the LLM to generate *[consequences]*, which reason about the future effects of possible actions. Then, we ask the LLM to generate *[affordance values]* of different object functional parts which indicate their usefulness in completing the given instruction.

Note that in this process, the *affordance prompting* technique is zero-shot in the sense that we do not provide any examples in the prompt to regulate the form of the output. This zero-shot usage makes the technique easily extend to different tasks. Empirically, the generated affordance values can effectively improve the feasibility of decomposed sub-tasks and control sequences from LLMs in the physical world.

To sum up, the *affordance prompting* technique provides the following advantages to facilitate reasoning in robotic tasks:

1. First, the *affordance prompting* technique assists LLMs in constraining control sequence updates within the set of feasible actions to follow the task instruction.
2. Secondly, the affordance values are convenient to obtain since only a single intermediate reasoning step is necessitated for LLMs, without requiring additional training or a fine-tuning process.
3. Thirdly, the *affordance prompting* technique adopts a zero-shot setting, which is robust across various envriments and capable of extending to heterogeneous tasks.

## 4 Experiments

We conduct experiments in various robotic tasks to answer the following questions: 1) How effective is LLM+A for physical interaction compared to other state-of-the-art baselines? 2) How well does LLM+A predict affordance values and plan action sequences? 3) How robust is LLM+A when generalized to heterogeneous tasks?

### 4.1 Experimental settings

**Implementation details.** For visual perception, given an image observation, we use the open-vocabulary detector OWL-ViT (Minderer et al., 2022) to detect the bounding boxes of relevant objects, and use GPT-4 (June) from OpenAI API (OpenAI, 2023) for both sub-task planner and motion controller. For the high-level sub-task planner, including the affordance prediction and consequence prediction, we don't provide any example outputs (zero-shot). For the low-level motion controller, we provide two examples in prompts to formalize the output style. The detailed prompts are listed in the Appendix A.1. Instead of using our LLM+A to re-plan new trajectory waypoints every time step, the plan will be updated after the robots finish the $K = 5$ waypoints in the previous round to increase the time efficiency.

**Tasks.** We evaluate LLM+A using four simulated task families from Language-Table (Lynch et al., 2023) and CLIPORT (Shridhar et al., 2022) as follows: 1) Block2Position: Push a block to an absolute location on the board (e.g., the top-left corner). The task is completed when the distance between the block and the target location is below a threshold. 2) Block2Block: Push a block to another block. The task is completed when the distance between the pushed block and the target block

Table 1: Success rates of LLM+A and baselines in pushing tasks from **Language-Table**.

| Method | Block2Position | Block2Block | Separate | Average |
|---|---|---|---|---|
| Naive LLM | 10% | 8% | 70% | 29% |
| ReAct | 15% | 3% | 40% | 19% |
| Code as Policies | 22% | 2% | 72% | 32% |
| LLM+A | **42%** | **32%** | **77%** | **50%** |

is below a threshold. 3) Separate: Separate two blocks. The task is completed when the distance between these blocks is below a threshold. 4) Pick&Palce: Place all blocks of a specified color into bowls of a specified color. The task is completed when all target blocks are within the bounds of specific bowls. The simulated environments in Task 1-3) employ an xArm6 robot, constrained to move in a 2D plane with a cylindrical end-effector. Task 4) utilizes a Universal Robot UR5e with a suction gripper. For each task, we evaluate our method and baselines for 100 episodes with random initial object positions and language instructions.

**Baselines.** We compare LLM+A with three baselines: 1) Naive LLM: Given the same inputs as LLM+A, we directly prompt the LLM to generate decomposed sub-task plans and motion plans without *affordance prompting*. 2) ReAct (Yao et al., 2023): An interactive decision-making approach of LLM by generating both reasoning traces and primitive actions in an interleaved manner. In our implementation, ReAct decides the best action in {Move Down, Move Up, Move Left, Move Right} with the robot coordinates and bounding boxes of blocks as inputs. By planning with such low-level actions, we evaluate its capability in motion control. 3) Code as Policies (Liang et al., 2023): An LLM-based code generation approach which directly calls pre-defined API to finish the task. In our implementation, after it translates the robotic tasks into programmatic codes, it finally calls one of the basic actions same as ReAct. For both ReAct and Code as Policies, we introduce two example cases in the prompt to help them understand the evaluation tasks. For all baselines, we use GPT-4 (June) as the base LLM. All specific prompts of baseline methods are in the Appendix A.2, A.3, and A.4.

## 4.2 RESULTS

**How effective is LLM+A compared to baselines?** We present the results of LLM+A and other baselines in three pushing tasks from Langage-Table in Table 1. LLM+A achieves the highest success rates and outperforms baseline methods significantly on Block2Position and Block2Block tasks. Our method achieves 50% average success rate across all tasks. In particular, we do not include any examples in the high-level sub-task planner when prompting GPT-4. This illustrates such physical knowledge is inherent in advanced LLMs and our approach can effectively stimulate this aspect of inference ability and make it valuable for robotics control tasks by *affordance prompting*. We show an example dialogue of the Block2Position task between LLM+A and GPT4 in Listing 6. When the task is to "push the cube to the top", the robot is commanded by LLM+A to approach the bottom side of the cube and then move it upwards. In contrast, we find that ReAct and Code as Policies only tend to imitate the procedure of the example outputs without truly understanding the physical consequences. Therefore, although we provide two examples for their prompts, the robot still makes mistakes in choosing the correct side. For example, when the task instruction is "move the cube to the upper left corner of the table", their methods guide the robot to approach the left side of the cube and push it towards the left direction. Their failure case examples are shown in Appendix A.6.

**How well does LLM+A predict affordance values and plan action sequences?** We display three examples of Block2Postion, Block2Block, and Separate tasks in Figure 3. First, we observe that our method can successfully predict goal-conditioned affordance values. For example, LLM assigns a higher affordance value to the bottom side A of the block to push it to the top side of the table as shown in Figure 3(a). Besides, the predicted affordance values also dynamically vary according to the current environment state. As shown in Figure 3(i), LLM firstly assigns a higher affordance score to the left side D of the red block at the beginning time. However, in Figure 3(k), LLM improves the affordance value to the top side G of the green block, since pushing side D could indirectly push the green block off the table. Second, our method can decompose the task instruction and generate the high-level current sub-task for the robot. As shown in Figure 3(f), the current sub-task is to push

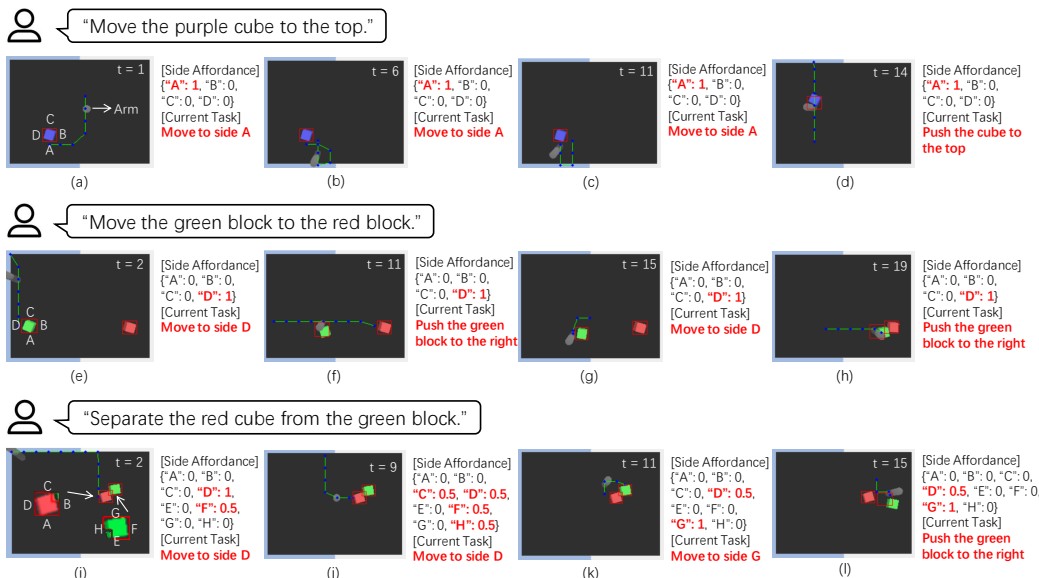

Figure 3: Examples of environmental observation and robot trajectories in Block2Position (a-d), Block2Block (e-h), and Separate (i-l). The gray cylinder indicates the position of the robot end-effector. The blue dots and the green lines represent the waypoints and the planned paths of the control sequences generated by LLM+A, respectively. The red boxes denote the detected bounding boxes from OWL-VIT.

Table 2: Success rates of LLM+A in Pick&Place tasks from **CLIPORT** with different numbers of target blocks.

| Number of blocks | 1-block | 2-blocks | 3-blocks | 4-blocks |
|---|---|---|---|---|
| Success Rate | 96% | 95% | 94% | 88% |

the green block to the right when the robot has approached side D. However, in Figure 3(g), LLM guides the robot to move to side D again when the robot breaks away from the block. Third, our method can generate executable low-level control sequences for the robot. For example, in Figure 3(c), LLM controls the robot to detour to side A of the purple block to prevent contact with the block in an unexpected direction. Owing to the low-level control sequences being raw coordinates, this proves that our approach is valid to arouse the spatial relationship understanding ability in LLMs, which is essential and promising for further non-training paradigms for robotics control.

**How robust is LLM+A when generalized to heterogeneous tasks?** We transfer LLM+A to Pick&Place tasks to evaluate the robustness of LLM+A on heterogeneous tasks. Notably, the prompts leveraged in the Pick&Place task are completely the same as in the pushing task except for the inputs of environmental information, which are shown in Appendix A.1. However, this task differs from pushing tasks in multiple aspects. First, the Pick&Place task leverages a Universal Robot UR5e different from the xArm6 robot in pushing tasks. Second, the interactive parts of blocks are block centers for the suction gripper different from multiple sides of blocks in pushing tasks for the cylindrical end-effctor. Third, the state dynamics of tabletop environments are different in these tasks. Therefore, LLMs need to adaptively understand the different physical interaction processes according to the robot type and task instructions. We variate the number of target blocks and report the success rates in Table 2. LLM+A achieves around 90% success rates in all scenarios. We show two trajectories of the Pick&Place task in Figure 4. Besides the target blocks and bowls, there are also many distractor objects on the table. Since multiple target blocks may exist, the robot needs to generate more sub-tasks to pick and place the blocks one by one and avoid putting multiple blocks into the same bowl. Our framework can be successfully transferred to this task. For example, the first example shows that the robot successfully recognizes four blue blocks and puts them into four

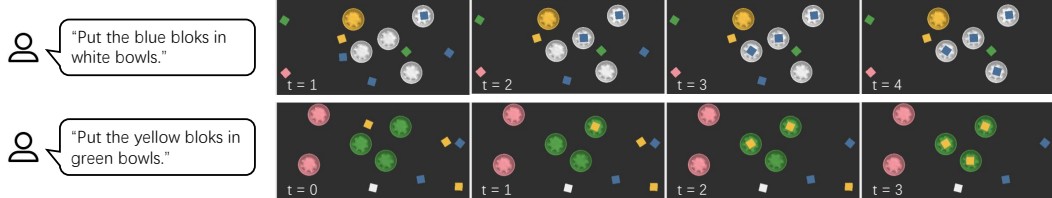

Figure 4: Examples of sequential environmental observation in the Pick&Place task. Given human language instruction, the robot needs to put specific blocks in target bowls. One bowl can only hold one block.

Table 3: Failure case analysis of LLM+A in fours tasks from **Language-Table** and **CLIPORT**.

| Failure | Block2Position | Block2Block | Separate | Pick&Place |
|---|---|---|---|---|
| Object Detection Fail | 2% | 6% | 6% | 12% |
| Affordance Prediction Fail | 12% | 20% | 10% | 0% |
| Task Planning Fail | 12% | 2% | 2% | 0% |
| Motion Controlling Fail | 16% | 22% | 4% | 0% |
| Exceed Time Budget | 16% | 18% | 1% | 0% |

different white bowls respectively. This suggests that *affordance prompting* is a general framework that can help the LLMs understand the attributes and consequences of robotic tasks, and make it generalizable for dealing with heterogeneous tasks. Besides, an example dialogue of LLM+A is shown in Listing 7.

### 4.3 FAILURE CASE ANALYSIS

Table 3 presents the failure case analysis of LLM+A in three pushing tasks from Langage-Table. Firstly, the object detection fail rates of the VLM are all lower than 6% in three tasks from Language-Table. Besides, we find that LLM fails more frequently in affordance prediction and motion controlling in Block2Block task. The main reason is that this task not only demands fine-grained motion plans for interacting with target blocks but also needs to consider the interactions between different blocks. Therefore, the planned control sequences from LLM can be further improved to avoid obstacles and reach the target as safely as possible. We will discuss this problem in our future work. Besides, we find that the object detection fail rate of Pick&Place task is higher than other tasks because VLM sometimes misjudges the color of the bowl. We can avoid this problem by using different suitable models for specific tasks.

## 5 CONCLUSION

In this paper, we study the language-conditioned robotic manipulation tasks with large models and propose the LLM+A framework, which successfully decomposes the language instruction into several sub-tasks, generates the robot control sequences, and easily extends to heterogeneous tasks. This shows the potential of LLMs in simultaneously achieving both planning and motion control, which provides an alternative training-free paradigm for utilizing the LLMs in robotic tasks. This significantly mitigates the dataset bottleneck issue for the robotics field. Besides, in order to ground the decomposed sub-tasks in the physical world and guarantee the feasibility of generated control sequences, we prove the proposed *affordance prompting* is crucial to stimulate the physical knowledge from the LLMs about spatial relationship and interaction consequences inference. The experimental results demonstrate the effectiveness of LLM+A. In the future, we will first optimize the process of using LLM+A to increase the time efficiency and secondly try to apply our method to more broad robotics tasks, including those tasks involving more complex physical interaction and different robot structures.

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

## A APPENDIX

### A.1 PROMPT FOR LLM+A

Listing 1: Prompt for LLM+A sub-task planner

```
1  [Guidelines]
2  You are an xArm6 robot (or a universal robot UR5e) on the tabletop,
3  constrained to move in a 2D plane with a cylindrical end-effector (or a
4  suction gripper). Your task is to finish language conditioned
5  manipulation task. The following coordinates are all from the top view.
6  Top left corner of the table is position [48, 4].
7  Top right corner of the table is position [271, 4].
8  Bottom left corner of the table is position [48, 172].
9  Bottom right corner of the table is position [271, 172].
10 The positive direction is right for x-axis and down for y-axis.
11 The smaller of 'x' means closer to the table left side,
12 the smaller of 'y' means closer to the table top side.
13 Remember, you do not need to return to the initial position after comple-
14 ting the task instruction.
15 [Inputs]
16 I will provide you with the 'Task Instruction', 'Arm Position', 'Object
17 Information', 'Possible Skills' in the JSON format.
18 {'Task Instruction': '',
19  'Arm Position': [,],
20  'Object Parts': {},
21  'Possible Skills': ['Push'](or ['Pick','Place'])}
22 [Outputs]
23 You should generate the following outputs in the JSON format:
24 {'Consequences': 'Imagine the consequences of the object states after
25 performing the 'possible skills' to different 'Object Parts'. ',
26 'Affordance':{'Based on the imagined 'Consequences', assign the
27 affordance value range from 0 to 1 for every part of the object to
28 represent their usefulness for the 'Task Instruction' with key 'Object
29 Parts'.'},
30 'Plan': [{'plan a sequence of sub-task with relative target coordinates
31 of your arm to finish the task with key 'sub_task' and
32 'target_coordinates'}]}
```

Listing 2: Prompt for LLM+A motion controller

```
1  [Guidelines]
2  You are an xArm6 robot (or a universal robot UR5e) on the tabletop,
3  constrained to move in a 2D plane with a cylindrical end-effector (or a
4  suction gripper). Your task is to finish language conditioned
5  manipulation task. The following coordinates are all from the top view.
6  Top left corner of the table is position [48, 4].
7  Top right corner of the table is position [271, 4].
8  Bottom left corner of the table is position [48, 172].
9  Bottom right corner of the table is position [271, 172].
10 The positive direction is right for x-axis and down for y-axis.
11 The smaller of 'x' means closer to the table left side,
12 the smaller of 'y' means closer to the table top side.
13 Given a 'Task Instruction', the 'Target Position', the current 'Arm
14 Position', and the 'Object Information' of the blocks on the table.
15 You need to generate the coordinate list of the planned path with key
16 'planned path' in the JSON format.
17 The maximum translation between two adjacent coordinate of the output
18 planned path is 20.
19 Firstly, I will give you two demos.
20 [Demo 1]
21 [Inputs]
22 {'Task Instruction': 'Move arm to the bottom side of the yellow cube.',
23 'Target Position':[92,118],
24 'Arm Position':[242,73],
```

```
25  'Object Information':{'blue cube':
26  {'side_top':{'coordinateA':[76,90],'coordinateB':[108,90]},
27  'side_left':{'coordinateA':[76,90],'coordinateB':[76,118]},
28  'side_bottom':{coordinateA':[76,118],'coordinateB':[108,118]},
29  'side_right':{'coordinateA':[108,118],'coordinateB':[108,90]},
30  'Affordance':{'side_top':0,'side_left':0,'side_bottom':1,
31  'side_right':0}}}}
32  [Outputs]
33  {"planned path":[[242,73],[242,93],[242,113],[222,118],[202,118],
34  [182,118],[162,118],[142,118],[122,118],[102,118],[92,118]]}
35  [Demo 2]
36  [Inputs]
37  {'Task Instruction': 'Push the blue cube to the top of the table.',
38  'Target Position':[92,4],
39  'Arm Position':[92,118],
40  'Object Information':{'green cube':
41  {'side_top':{'coordinateA':[76,90],'coordinateB':[108,90]},
42  'side_left':{'coordinateA':[76,90],'coordinateB':[76,118]},
43  'side_bottom':{coordinateA':[76,118],'coordinateB':[108,118]},
44  'side_right':{'coordinateA':[108,118],'coordinateB':[108,90]},
45  'Affordance':{'side_top':0,'side_left':0,'side_bottom':1,
46  'side_right':0}}}}
47  [Outputs]
48  {"planned path":[[92,118],[92,98],[92,78],[92,58],[92,38],
49  [92,18],[92,4]]}
```

## A.2 PROMPT FOR NAIVE LLM

Listing 3: Prompt for Naive LLM sub-task planner

```
1   [Guidelines]
2   You are an xArm6 robot (or a universal robot UR5e) on the tabletop,
3   constrained to move in a 2D plane with a cylindrical end-effector (or a
4   suction gripper). Your task is to finish language conditioned
5   manipulation task. The following coordinates are all from the top view.
6   Top left corner of the table is position [48, 4].
7   Top right corner of the table is position [271, 4].
8   Bottom left corner of the table is position [48, 172].
9   Bottom right corner of the table is position [271, 172].
10  The positive direction is right for x-axis and down for y-axis.
11  The smaller of 'x' means closer to the table left side,
12  the smaller of 'y' means closer to the table top side.
13  Remember, you do not need to return to the initial position after comple-
14  ting the task instruction.
15  [Inputs]
16  I will provide you with the 'Task Instruction', 'Arm Position', 'Object
17  Information', 'Possible Skills' in the JSON format.
18  {'Task Instruction': '',
19   'Arm Position': [,],
20   'Object Parts': {},
21   'Possible Skills': ['Push'](or ['Pick','Place'])}  }
22  [Outputs]
23  You should generate the following outputs in the JSON format:
24  {'Plan': [{'plan a sequence of sub-task with relative target coordinates
25   of your arm to finish the task with key 'sub_task' and
26   'target_coordinates'}]}
```

## A.3 PROMPT FOR REACT

Listing 4: Example Prompt for ReAct

```
1   You are a robot arm working on the tabletop. Looking quickly around you,
2   you see your arm at position [x,y]=[60,50], a ['green block'] with bound-
```

```
3   ing box coordinates [[x1,y1,x2,y2]] = [[90,90,110,110]]. Your task is to:
4   slide the green block to the bottom. The bounding box of tabletop coordi-
5   nates [x1,y1,x2,y2] = [0,0,200,150]. The greater in 'x' coordinate means
6   more right, the greater in 'y' coordinates means more bottom.
7   Act 1: Think<To push the green block to the bottom, I should approach the
8   top edge of the block. The top edge coordinates is [90,90,110,90]. And my
9   arm's x coordinate 60 is less than edge's coordinate x1=90, and y coordi-
10  nate 50 is also less than edge's coordinates y1=90, so I am at the top-
11  left of the top edge. To approach the top edge, I should move right.>
12  Act 2: Move Right
13  Obs 2: Looking quickly around you, you see your arm at position [x,y]=
14  [80,50], and a ['green block'] with bounding box coordinates = [[90,90,
15  110,110]].
16  Act 3: Think<To push the green block to the bottom, I should approach the
17  top edge of the block. The top edge coordinates is [90,90,110,90]. And my
18  arm's x coordinate 80 is less than edge's coordinate x1=90 and y coordi-
19  nate 50 is also less than edge's coordinates y1=90, so I am still at the
20  top-left of the top edge. To approach the top edge, I should move right.>
21  Act 4: Move Right
22  Obs 4: Looking quickly around you, you see your arm at position [x,y]=
23  [100,50], and a ['green block'] with bounding box coordinates = [[90,90,
24  110,110]].
25  Act 5: Think<To push the green block to the bottom, I should approach the
26  top edge of the block. The top edge coordinates is [90,90,110,90]. My arm
27  's x coordinate 100 is greater than the edge's coordinate x1=90 and less
28  than x3=110, y coordinate less than edge's coordinate y1=90, so I am at
29  the top of the top edge. To approach the top edge, I should move down.>
30  Act 6: Move Down
31  Obs 6: Looking quickly around you, you see your arm at position [x,y]=
32  [100,70], and a ['green block'] with bounding box coordinates = [[90,90,
33  110,110]].
34  Act 7: Think<To push the green block to the bottom, I should approach the
35  top edge of the block. The top edge coordinates is [90,90,110,90]. My arm
36  's x coordinate 100 is greater than the edge's coordinate x1=90 and less
37  than x3=110, y coordinate less than edge's coordinate y1=90, so I am at
38  the top of the top edge. To approach the top edge, I should move down.>
39  Act 8: Move Down
40  Obs 8: Looking quickly around you, you see your arm at position [x,y]=
41  [100,90], and a ['green block'] with bounding box coordinates = [[90,90,
42  110,110]].
43  Act 9: Think<To push the green block to the bottom, I should approach the
44  top edge of the block. The top edge coordinates is [90,90,110,90]. And my
45  arm's x coordinate 100 is greater than the edge's coordinate x1=90 and
46  less than x3=110, y coordinate is close to the edge's coordinate y1=90,
47  so I am very closed to the top edge and I can push it.>
48  Act 10: Push Down
49  Obs 10: Looking quickly around you, you see your arm at position [x,y]=
50  [100,110], and a ['green block'] with bounding box coordinates = [[90,
51  115,110,135]].
52  Act 11: Think<To push the green block to the bottom, I should approach
53  the top edge of the block. The top edge coordinates is [90,90,110,90].
54  And my arm's x coordinate 100 is greater than the edge's coordinate x1=90
55  and less than x3=110, y coordinate is close to 90, so I am very closed to
56  the top edge and I can push it.>
57  Act 12: Push Down
58  Obs 12: Looking quickly around you, you see your arm at position [x,y]=
59  [100,130], and a ['green block'] with bounding box coordinates = [[90,
60  135,110,155]].
61  Act 13: Think<Based on the coordinates, the block has reached the bottom
62  side with coordinates y=150, so I finished the task.>
63  Act 14: Succeed
```

## A.4 PROMPT FOR CODE AS POLICIES

Listing 5: Example Prompt for Code as Policies

```python
# Python 2D robot control script
import numpy as np
from env_utils import get_obs_image,get_robot_pose, get_obj_box,
get_obj_edge, push, move, say, detect_relation, detect_arrive_side
# push the yellow block to the bottom right corner
say('Ok - pushing the yellow block to the bottom right corner')
image = get_obs_image(observation)
robot = get_robot_pose(observation)
yellow_block_xyxy = get_obj_box(observation,'yellow_block')
say("Robot pose={}, Block pose={}".format(robot,yellow_block_xyxy))
yellow_block_top_edge = get_obj_edge(yellow_block_xyxy,'top_side')
yellow_block_left_edge = get_obj_edge(yellow_block_xyxy,'left_side')
top_relation = detect_relation(robot,yellow_block_top_edge,'top')
left_relation = detect_relation(robot,yellow_block_left_edge,'left')
bottom_side_xyxy = get_side_pose('bottom_side')
right_side_xyxy = get_side_pose('right_side')
arrive_bottom_side = detect_arrive_side('bottom_side', bottom_side_xyxy,
 yellow_block_xyxy)
arrive_right_side = detect_arrive_side('right_side', right_side_xyxy,
 yellow_block_xyxy)
if not arrive_bottom_side:
    if top_relation == "arrive":
        push(robot,'down')
        say("push down")
    elif top_relation == "top":
        move(robot,'down')
        say("move down")
    elif top_relation == "top_left":
        move(robot,'right')
        say("move right")
    elif top_relation == "top_right":
        move(robot,'left')
        say("move left")
    elif top_relation == "bottom":
        move(robot,'left')
        say("move left")
    elif top_relation == "bottom_left":
        move(robot,'up')
        say("move up")
    elif top_relation == "bottom_right":
        move(robot,'up')
        say("move up")
elif not arrive_right_side:
    if left_relation == "arrive":
        push(robot,'right')
        say("push right")
    elif left_relation == "left":
        move(robot,'right')
        say("move right")
    elif left_relation == "top_left":
        move(robot,'down')
        say("move down")
    elif left_relation == "bottom_left":
        move(robot,'up')
        say("move up")
    elif left_relation == "right":
        move(robot,'down')
        say("move down")
    elif left_relation == "bottom_right":
        move(robot,'left')
        say("move left")
```

```
62        elif left_relation == "top_right":
63            move(robot,'left')
64            say("move_left")
```

## A.5    EXAMPLE SUCCESSES

Listing 6: Example success of LLM+A sub-task planner in Block2Position task

```
1   [Block2Position Task]
2   You are an xArm6 robot on the tabletop,
3   constrained to move in a 2D plane with a cylindrical end-effector (or a
4   suction gripper). Your task is to finish language conditioned
5   manipulation task. The following coordinates are all from the top view.
6   Top left corner of the table is position [48, 4].
7   Top right corner of the table is position [271, 4].
8   Bottom left corner of the table is position [48, 172].
9   Bottom right corner of the table is position [271, 172].
10  The positive direction is right for x-axis and down for y-axis.
11  The smaller of 'x' means closer to the table left side,
12  the smaller of 'y' means closer to the table top side.
13  Remember, you do not need to return to the initial position after comple-
14  ting the task instruction.
15  I will provide you with the 'Task Instruction', 'Arm Position', 'Object
16  Information', 'Possible Skills' in the JSON format.
17  {'Task Instruction': 'move the green cube to the top',
18   'Arm Position': [127,83],
19   'Object Parts': {'green cube':{
20   'edge_top':{'coordinateA':[110, 57],'coordinateB':[144, 57]},
21   'edge_left':{'coordinateA':[110, 57],'coordinateB':[110, 88]},
22   'edge_bottom':{'coordinateA':[110, 88],'coordinateB':[144,88]},
23   'edge_right':{'coordinateA':[144, 88],'coordinateB':[144, 57]}}}},
24   'Possible Skills': ['Push']}
25  You should generate the following outputs in the JSON format:
26  {'Consequences': 'Imagine the consequences of the object states after
27  performing the 'possible skills' to different 'Object Parts'. ',
28  'Affordance':{'Based on the imagined 'Consequences', assign the
29  affordance value range from 0 to 1 for every part of the object to
30  represent their usefulness for the 'Task Instruction' with key 'Object
31  Parts'.'},
32  'Plan': [{'plan a sequence of sub-task with relative target coordinates
33  of your arm to finish the task with key 'sub_task' and
34  'target_coordinates'}]}
35
36  [GPT4]
37  {'Consequences': {
38  'push_edge_top': 'The blue cube will move downwards.',
39  'push_edge_left': 'The blue cube will move to the right.',
40  'push_edge_bottom': 'The blue cube will move upwards.',
41  'push_edge_right': 'The blue cube will move to the left.'},
42  'Affordance': {'edge_top': 0,'edge_left': 0,
43  'edge_bottom': 1,'edge_right': 0},
44  'Plan': [{
45  'sub_task': 'Move arm to edge_bottom of the blue cube',
46  'target_coordinates': [125, 130]},
47  {'sub_task': 'Push the blue cube to the top of the table',
48  'target_coordinates': [125, 4]}]}
```

Listing 7: Example success of LLM+A sub-task planner in Pick&Place task

```
1   [Pick&Place Task]
2   You are an xArm6 robot on the tabletop,
3   constrained to move in a 2D plane with a cylindrical end-effector (or a
4   suction gripper). Your task is to finish language conditioned
```

```
 5  manipulation task. The following coordinates are all from the top view.
 6  Top left corner of the table is position [48, 4].
 7  Top right corner of the table is position [271, 4].
 8  Bottom left corner of the table is position [48, 172].
 9  Bottom right corner of the table is position [271, 172].
10  The positive direction is right for x-axis and down for y-axis.
11  The smaller of 'x' means closer to the table left side,
12  the smaller of 'y' means closer to the table top side.
13  Remember, you do not need to return to the initial position after comple-
14  ting the task instruction.
15  I will provide you with the 'Task Instruction', 'Object Information',
16  'Possible Skills' in the JSON format.
17  {'Task Instruction': 'put the pink blocks in a white bowl',
18   'Arm Position': [127,83],
19   'Object Parts': [
20  {'object id':5, 'type': 'white bowl', 'pose': [220, 301, 0]},
21  {'object id':6, 'type': 'white bowl', 'pose': [123, 246, 0]},
22  {'object id':7, 'type': 'white bowl', 'pose': [488, 162, 0]},
23  {'object id':8, 'type': 'pink block', 'pose': [139, 239, 0]},
24  {'object id':9, 'type': 'pink block', 'pose': [462, 107, 0]},
25  {'object id':10, 'type': 'blue block', 'pose': [254, 221, 0]},
26  {'object id':11, 'type': 'red block', 'pose': [109, 107, 0]},
27  {'object id':12, 'type': 'green bowl', 'pose': [586, 285, 0]},
28  {'object id':13, 'type': 'yellow bowl', 'pose': [444, 300, 0]},
29  {'object id':14, 'type': 'blue block', 'pose': [385, 136, 0]}],
30   'Possible Skills': ['Pick', 'Place']}
31  You should generate the following outputs in the JSON format:
32  {'Consequences': 'Imagine the consequences of the object states after
33  performing the 'possible skills' to different 'Object Parts'. ',
34  'Affordance':{'Based on the imagined 'Consequences', assign the
35  affordance value range from 0 to 1 for every part of the object to
36  represent their usefulness for the 'Task Instruction' with key 'Object
37  Parts'.'},
38  'Plan': [{'plan a sequence of sub-task with relative target coordinates
39  of your arm to finish the task with key 'sub_task', 'possible skills' and
40  'target_coordinates'}]}
41
42  [GPT4]
43  {'Consequences': [
44  {'object id': 5, 'state': 'white bowl with pink block'},
45  {'object id': 6, 'state': 'white bowl with pink block'},
46  {'object id': 7, 'state': 'empty white bowl'},
47  {'object id': 8, 'state': 'pink block in white bowl'},
48  {'object id': 9, 'state': 'pink block in white bowl'},
49  {'object id': 10, 'state': 'blue block'},
50  {'object id': 11, 'state': 'red block'},
51  {'object id': 12, 'state': 'green bowl'},
52  {'object id': 13, 'state': 'yellow bowl'},
53  {'object id': 14, 'state': 'blue block'}],
54  'Affordance': {'5': 1, '6': 1, '7': 0, '8': 1, '9': 1,'10': 0,
55  '11': 0,'12': 0,'13': 0,'14': 0},
56  'Plan': [
57  {'sub_task': 'Pick up the first pink block',
58  'possible skills': 'pick',
59  'target_coordinates': [139, 239]},
60  {'sub_task': 'Place the first pink block in the first white bowl',
61  'possible skills': 'place',
62  'target_coordinates': [220, 301]},
63  {'sub_task': 'Pick up the second pink block',
64  'possible skills': 'pick',
65  'target_coordinates': [462, 107]},
66  {'sub_task': 'Place the second pink block in the second white bowl',
67  'possible skills': 'place',
68  'target_coordinates': [123, 246]}]}}
```

## A.6 EXAMPLE FAILURES

Listing 8: Failure case for Naive LLM, the red fonts represents the mistakes in thinking. We found that Naive LLM makes many mistakes in spatial relationship reasoning. Specifically, naive LLM first fails in choosing the skills and proposes the wrong target coordinates in the subsequent step.

```
[Block2Position Task]
You are an xArm6 robot on the tabletop,
constrained to move in a 2D plane with a cylindrical end-effector (or a
suction gripper). Your task is to finish language conditioned
manipulation task. The following coordinates are all from the top view.
Top left corner of the table is position [48, 4].
Top right corner of the table is position [271, 4].
Bottom left corner of the table is position [48, 172].
Bottom right corner of the table is position [271, 172].
The positive direction is right for x-axis and down for y-axis.
The smaller of 'x' means closer to the table left side,
the smaller of 'y' means closer to the table top side.
Remember, you do not need to return to the initial position after comple-
ting the task instruction.
I will provide you with the 'Task Instruction', 'Arm Position', 'Object
Information', 'Possible Skills' in the JSON format.
{'Task Instruction': 'move the green cube to the top',
 'Arm Position': [127,83],
 'Object Parts': {'green cube':{
 'edge_top':{'coordinateA':[110, 57],'coordinateB':[144, 57]},
 'edge_left':{'coordinateA':[110, 57],'coordinateB':[110, 88]},
 'edge_bottom':{'coordinateA':[110, 88],'coordinateB':[144,88]},
 'edge_right':{'coordinateA':[144, 88],'coordinateB':[144, 57]}}},
 'Possible Skills': ['Push']}
You should generate the following outputs in the JSON format:
{'Plan': [{'plan a sequence of sub-task with relative target coordinates
of your arm to finish the task with key 'sub_task' and
'target_coordinates'}]}

[GPT4]
{ "Plan": [{
"sub_task": "Move to the green cube", "target_coordinates": [127,57]},
{"sub_task": "Grab the green cube", "target_coordinates": [127,57]},
{"sub_task": "Move to the top", "target_coordinates": [127,4]},
{"sub_task": "Release the green cube", "target_coordinates": [127,4]}]}
```

To align the ReAct approach to our tasks, we follow the rules in the original paper and design new prompt examples for interleaved reasoning and action as shown in 4. For the thinking process, we hint to the LLM about how to infer the spatial relationships and the optimal actions. We found that the ReAct can make reasonable thinking, (e.g. after performing the pushing skills, if the block doesn't move, it knows to move to another side), but it will also make mistakes on the physical interaction inference. An examples is shown in 9. In order to move the green cube to the upper left corner, it should approach the right side instead of the left edge. And once a mistake occurs, such a thinking process will influence all the subsequent decisions, which makes it fail in most of the tasks.

Listing 9: Failure case for ReAct, the red fonts represents the mistakes in thinking, the orange fonts represents the correct answer. We found that ReAct makes many mistakes in spatial relationship inference and interaction consequence prediction.

```
Act 49: Think<To move the green cube to the upper left corner, I should
approach the left edge (right edge) of the cube. Based on the coordinates
my 'x' is less thans the cube 'x1' and my 'y' is less than the cube 'y1',
so I am still at the bottom-left (top-left) of the green cube, to appro-
ach the left edge, I should move right.>
```

We provide GPT-4 with two example codes as in 5 for each task to understand the protocol of pushing the blocks. To evaluate the capability of the LLM in low-level motion planning, we regulate the

LLM can only select the API of moving/pushing up, down, left, or right. From our experiments, the code-like prompts allow the LLM to own outstanding task decomposition ability, and generalization across various instruction inputs, but it still makes mistakes in choosing the correct side for finishing the pushing block tasks. An example is shown in 10. In other words, such methods don't inspire the physical grounding abilities of LLM. And for many tasks, especially those requiring low-level motion planning tasks, it is difficult to explicitly summarize the whole task into a step-to-step procedure as code. But our approach provides a new paradigm.

Listing 10: Failure case of code as policies. The mistake is marked with red fonts. Although the entire procedure is correct, it sometimes fails in finding the correspodning edge for completing the task. And for local motion-planning, it is complicated or even infeasible to write the step-to-step procedure as code.

```
say("Ok - moving the yellow block to the left side of the board")
image = get_obs_image(observation)
robot = get_robot_pose(observation)
yellow_block_xyxy = get_obj_box(observation,"yellow block")
yellow_block_left_edge = get_obj_edge(yellow_block_xyxy,"left side")
left_relation = detect_relation(robot,yellow_block_left_edge,"left")
left_side_xyxy = get_side_pose("left_side")
arrive_left_side = detect_arrive_side("left_side",left_side_xyxy,
    yellow_block_xyxy)
if not arrive_left_side:
    if left_relation == "arrive":
        push(robot,"left")
        say("push left")
    elif left_relation == "left":
        move(robot,"left")
        say("move left")
    elif left_relation == "top_left":
        move(robot,"down")
        say("move down")
    elif left_relation == "bottom_left":
        move(robot,"up")
        say("move up")
    elif left_relation == "right":
        move(robot,"down")
        say("move down")
    elif left_relation == "bottom_right":
        move(robot,"left")
        say("move left")
    elif left_relation == "top_right":
        move(robot,"left")
        say("move left")
```

## A.7 GENERALIZATION TO BLOCKS WITH DIFFERENT SHAPES

To evaluate the robustness of our proposed LLM+A method, we further conduct one robotics control task (Block2Position) with respect to four types of blocks with different shapes (cube, star, pentagon, moon). All the settings are the same except the block shapes. We evaluate 100 episodes for this task, and our LLM+A achieves 38% average success rate, which is consistent to the performance achieved on cubes. Note that using the bounding box as input may contradict to the shape of the block and cause unpredictable physical interaction results. However, as our method control the locomotion in close-looped manner, it will dynamically adjust its planning trajectory. That is the main reason of why using bounding box is applicable. The examples are shown in Fig 5.

## A.8 GENERALIZATION TO BLOCKS WITH ENVIRONMENT INTERFERENCE

In the real-world robotics applications, the objects placement are easily influenced by our human activities. To verify whether our method can deal with such scenarios, we randomly change the block's position as human interference in Block2Position task to see whether the LLM+A can adjust

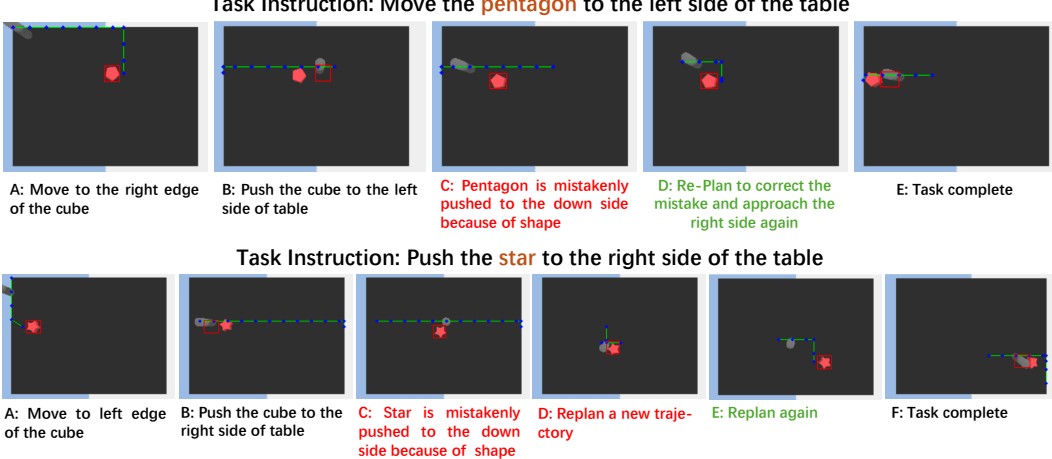

Figure 5: Trajectory Visualization. We demonstrate two examples to illustrate the re-planning process. As shown in figures, the unpredictable interactions will happen and our method can successfully change its plan considering the location feedback of the objects.

its plan in time. We evaluate for another 100 episode and our method achieves 40% in success rate, which is consistent to the main results. The examples are shown in Fig6.

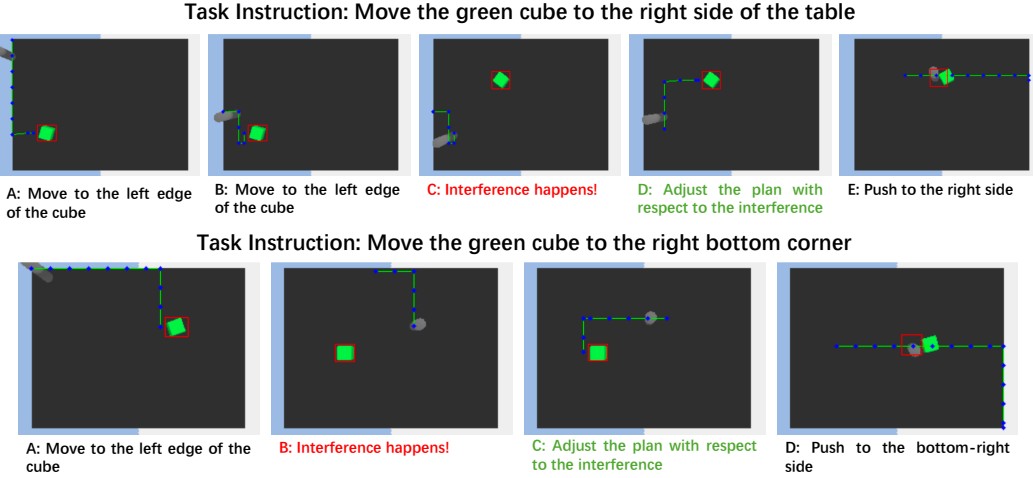

Figure 6: Trajectory Visualization. We demonstrate two examples to illustrate the re-planning process with respect to human interference. As shown in figures, once the block is taken far away, the planned trajectory will be immediately re-planned to catch up with the blocks.

