# OpenReview forum: "LLM+A: Grounding Large Language Models in Physical World with Affordance Prompting"
_ICLR.cc/2024/Conference — Submitted to ICLR 2024_

### Official Review · Reviewer_xvo1 · 2023-10-27

**Soundness:** 2 fair
**Presentation:** 3 good
**Contribution:** 2 fair
**Rating:** 5
**Confidence:** 4

**Summary:**

(1) This paper studied language-conditioned robotic manipulation tasks using large models, proposing LLM+A framework that can decompose language instructions into sub-tasks.

(2) Generated robot control sequences and extended to heterogeneous tasks and demonstrated potential of LLMs in planning and motion control simultaneously

(3) Provided training-free paradigm for utilizing LLMs in robotic tasks, addressing dataset bottleneck and Affirmed the importance of affordance prompting for grounding sub-tasks in physical world. Experiments proved the effectiveness of LLM+A
Planned future optimizations for time efficiency and application to complex robotics tasks

**Strengths:**

(1) This work integrate LLMs into robot planning and reasoning. It leverages recent advancements in LLMs and it utilizes LLMs as high-level sub-task planners and low-level motion controllers in robotic tasks.

(2) Several challenges are mentioned and addressed, including identifying the pre-trained skills and sub-policies, and generalizing unseen envs and diverse scenes.

(3) While introducing LLM + A framework, this work enhanced robotic manipulation tasks by grounding LLMs in the physical world, improving motion plans by considering affordance knowledge.

**Weaknesses:**

(1) [Limited Generalization]: The experiment of this study depends on the simplified tasks, such as pushing cubes and put cubes. These objects are rather simple with regular shapes. This is crucial to discretize the actions. However,  the experiments may not cover all possible scenarios with more complex envs and irregular daily objects, leading to potential limitations in the model's applicability outside the tested conditions.

(2) [Affordance Predictions Accuracy] The accuracy of affordance predictions from LLMs could be a potential weakness. If the affordance values are inaccurately predicted, it might lead to sub-optimal or erroneous robotic actions. Variability of landscapes, backgrounds, and language descriptions predicting affordances across different objects or environments could impact the overall performance.

Overall, the experiments might lack certain real-world complexities, such as dynamic and unpredictable environments. For example, the Owl-vit vision grounding module only predicts the bounding box. Which is not ideal for most of the cases in real-world applications.

**Questions:**

(1) What is the inference speed of the planning? Prompting GPT-4 takes some time to response compared with other simpler models e.g. from Ros integration.

(2) Why not implement SAM based vision detection modules for more accurate detection and generalize the tasks into more complex scenarios.

**Details Of Ethics Concerns:**

I don't see any ethical concerns since it is the embodied robotic study.

---

> ### Author Response · Authors · 2023-11-22
> **Response to Reviewer xvo1 (1/2)**
>
> Dear Reviewer xvo1,
>
> We greatly appreciate your insightful comments and the time you took to review our manuscript. Your feedback is essential to improving our work, and here we address your concerns:
> > 1. [Limited Generalization]: The experiment of this study depends on the simplified tasks, such as pushing cubes and put cubes. These objects are rather simple with regular shapes. This is crucial to discretize the actions. However, the experiments may not cover all possible scenarios with more complex envs and irregular daily objects, leading to potential limitations in the model's applicability outside the tested conditions.
>
> Thank you for your valuable suggestions. We agree with you that the experiment should include more complex tasks to verify the generalization ability of our method. Accordingly, we have incorporated additional testing scenarios. First, we diversified the shape of objects, introducing cubes, pentagons, star-shaped, and moon-shaped objects. We conducted an evaluation of our method over 100 episodes in the Block2Position task, resulting in a success rate of 38%. Notably, this rate is nearly consistent with the success rate reported in our paper. An example is shown in Figure 5. Second, we applied our method in complex scenarios, involving potential interferences with object positions. More specifically, we randomly change the object position at any given time step within an episode. An example of environmental observation and robot trajectories is provided in Figure 6 for your reference. The results demonstrate the robustness of our method, notably its ability to dynamically recalibrate motion sequences.
>
> > 2. [Affordance Predictions Accuracy] The accuracy of affordance predictions from LLMs could be a potential weakness. If the affordance values are inaccurately predicted, it might lead to sub-optimal or erroneous robotic actions. Variability of landscapes, backgrounds, and language descriptions predicting affordances across different objects or environments could impact the overall performance.
>
> We agree with your concern about the affordance prediction accuracy. Variability of landscapes, backgrounds, and language descriptions across different objects or environments could impact the over-performance. In our experiments, the task instruction has no pre-defined template, grammar, or vocabulary for each task. For instance, “move the red block to the top right of the board” can also be described as “push the red block to the upper right corner”. The experiment results show that LLM is robust to the variability of language descriptions.
>
> Besides, the robustness to the variability of landscapes and backgrounds is mainly dependent on the capability of detection models, recent visual-language models can achieve pleasant detection/segmentation results with respect to text inputs, and the accuracy of the perception module is not the focus of our work.
>
> Furthermore, even if the affordance prediction can make mistakes sometimes, as our method takes in the detection results as the planner’s input which composes the closed-loop control, it can greatly alleviate the planning prediction error in one round by consistently adjusting its planning trajectory.

---

> > ### Author Response · Authors · 2023-11-22
> > **esponse to Reviewer xvo1 (2/2)**
> >
> > > Overall, the experiments might lack certain real-world complexities, such as dynamic and unpredictable environments. For example, the Owl-vit vision grounding module only predicts the bounding box. Which is not ideal for most of the cases in real-world applications.
> >
> > Thank you for your comments, and I agree with you that real-world experiments are the best way to illustrate LLM+A’s ability in robotics control. And we will append the real-world experiments in our future work. But in principle, as our method works in a zero-shot manner, it is less likely to suffer from the sim-to-real transfer. First of all, the inputs of our LLM+A system only include the bounding box of the objects. Recent advances in general visual grounding modules/visual-language models can achieve satisfying accuracy in predicting the bounding box/segmentation mask of the objects, even for object parts [1][2][3]. The RGB-D camera is commonly used in robotics manipulation and grasping fields. With the help of the depth sensor, it is straightforward to translate the planned waypoints from pixel space into a 3-D coordinates in the real world, and then use IK to control the locomotion of the robot arm. I understand your concern about the generalization of the bounding-box-based inputs. However, our framework focuses on the “parts” of the objects instead of being limited in bounding boxes. In our experiments, the simulator provides tabletop manipulation tasks on blocks, so it is natural to use bounding boxes and edges to represent the “parts” of a block. But for real-world applications, an object often comes with functional parts, like the hammer is composed of a handle and a hammer head. It is not difficult for the LLM to first infer the parts of the objects and then use vision-language models to predict the location of different parts to serve as our LLM+A inputs.
> >
> > [1] Kirillov, A., Mintun, E., Ravi, N., Mao, H., Rolland, C., Gustafson, L., Xiao, T., Whitehead, S., Berg, A.C., Lo, W., Dollár, P., & Girshick, R.B. (2023). Segment Anything. ArXiv, abs/2304.02643.
> >
> > [2] Lai, X., Tian, Z., Chen, Y., Li, Y., Yuan, Y., Liu, S., & Jia, J. (2023). LISA: Reasoning Segmentation via Large Language Model. ArXiv, abs/2308.00692.
> >
> > [3] Wang, W., Chen, Z., Chen, X., Wu, J., Zhu, X., Zeng, G., Luo, P., Lu, T., Zhou, J., Qiao, Y., & Dai, J. (2023). VisionLLM: Large Language Model is also an Open-Ended Decoder for Vision-Centric Tasks. ArXiv, abs/2305.11175.
> >
> > > What is the inference speed of the planning? Prompting GPT-4 takes some time to response compared with other simpler models e.g. from Ros integration.
> >
> > To increase the time efficiency, we query GPT4 to in a fixed interval generate sub-task plans and motion sequences instead of re-planning new trajectory waypoints every time step. Specifically, the planned sequence will be updated only after the robot finishes five waypoints in the previous query round. The average query number of each episode in evaluation tasks can be referred to in the following table. We do not include the query time since it depends on different sources of OpenAI APIs.
> > | Task | Block2Position | Block2Block | Separate |
> > | --- | --- | --- | ---|
> > | GPT4 average query number | 3.86 | 3.75 | 1.82 |
> >
> > > Why not implement SAM based vision detection modules for more accurate detection and generalize the tasks into more complex scenarios.
> >
> > Thank you for your advice. SAM can provide a more detailed description of the object's shape. However such detailed information is difficult to translate into text-inputs for LLM to understand. Maybe the SAM mask input is more suitable for  VLM (like recent GPT-4V) to do the plan. We use the bounding-box detection inputs as a trade-off between the description details and the difficulty of translating into texts. One major disadvantage of using the bounding box is that the object will not occupy all the space indicated by the bounding box. The planner may plan a waypoint that does not actually lead to interactions between the robot arm and the object. But as our LLM+A uses the detection results as feedback, it will be aware of such mistakes and improve its previously planned trajectory to fix this.

---

### Official Review · Reviewer_qNCi · 2023-10-30

**Soundness:** 2 fair
**Presentation:** 3 good
**Contribution:** 2 fair
**Rating:** 3
**Confidence:** 5

**Summary:**

This paper provides a prompt framework to let LLM solve robotics tasks w/o any training. The main innovation is that it forces the LLM to output affordance that is a constraint to make sure the control is within the set of feasible actions to follow the task instruction. Experimental results show that it is better than code as policies and ReAct for the tasks included in this paper.

**Strengths:**

The idea of constraining the LLM to output action according to affordance is interesting.

**Weaknesses:**

1. The paper title claims physical world, however, in the evaluation, it only considers simulation tasks on table top (2D). Physical world robotics interaction is much more complicated than simulation and will absolutely break the assumption of this paper. In my point of view, the technique proposed in this paper only applies to a very limited setup. Basically, given some 2D points (target positions) how to use robot arm (source positions) to reach it and generate some trajectories. In contrary, techniques such as code as policies is general and can extend to physical world.

2. The technique proposed by this paper may highly depend on the choice of LLM. Ablations w/ different LLMs is required to show generality.

3. In SayCan paper, there is an "open source environment" section. Looks like the tasks are similar. It will be interesting to see the comparison w/ SayCan.

**Questions:**

Is the policy able to re-try if the first trial fails?

---

> ### Author Response · Authors · 2023-11-22
> **Response to Reviewer qNCi**
>
> Dear Reviewer qNCi,
>
> We greatly appreciate your insightful comments and the time you took to review our manuscript. Your feedback is essential to improving our work, and here we address your concerns:
> > 1. The paper title claims physical world, however, in the evaluation, it only considers simulation tasks on table top (2D). Physical world robotics interaction is much more complicated than simulation and will absolutely break the assumption of this paper. In my point of view, the technique proposed in this paper only applies to a very limited setup. Basically, given some 2D points (target positions) how to use robot arm (source positions) to reach it and generate some trajectories. In contrary, techniques such as code as policies is general and can extend to physical world.
>
> Thank you for your comments. In [1], the simulated environment “Language-Table” consists of a simulated 6DoF robot xArm6 implemented in PyBullet equipped with a small cylindrical end effector. Third-person perspective RGB-only images from a simulated camera are used as visual input. Also, similar real-world experiments are tested using UFACTORY xArm6 robot arms. The authors have demonstrated that the policy performance in Language-Table was highly correlated with policy performance in the real world. In [2], the Pick&Place task is built based on the Ravens benchmark [3] set in PyBullet. Although the rendering software stack may not fully reflect the noise characteristics often found in real data, the simulated environment can roughly match the real-world setup. Despite the 2D environment, we can transfer our method into a 3D setup using the following pipeline: we first invoke open-vocab detector OWL-ViT [4] to obtain a bounding box, and feed it into Segment Anything [5] to obtain an object mask. Then the object point cloud can be reconstructed with the mask and the RGB-D observation.
>
> [1] Lynch, Corey, et al. "Interactive language: Talking to robots in real time." IEEE Robotics and Automation Letters (2023).
>
> [2] Shridhar, Mohit, Lucas Manuelli, and Dieter Fox. "Cliport: What and where pathways for robotic manipulation." Conference on Robot Learning. PMLR, 2022.
>
> [3] Zeng, Andy, et al. "Transporter networks: Rearranging the visual world for robotic manipulation." Conference on Robot Learning. PMLR, 2021.
>
> [4] Minderer, M., et al. "Simple open-vocabulary object detection with vision transformers”. arXiv 2022. arXiv preprint arXiv:2205.06230.
>
> [5] Kirillov, Alexander, et al. "Segment anything." arXiv preprint arXiv:2304.02643 (2023).
>
> > 2. The technique proposed by this paper may highly depend on the choice of LLM. Ablations w/ different LLMs is required to show generality.
>
> Thank you for your valuable suggestion. Our method currently still depends on the powerful GPT4. GPT4 is prone to generate outputs as we instructed that can be executed in the code, which is necessary for the manipulator policy. However, open-source models like Llama2 or Vicuna often throw errors both syntactically and semantically even if we show exemplars and instruct the output format in the prompt. Maybe instruction finetuning can alleviate this issue, but this is not the focus of our paper.
>
> > 3. In SayCan paper, there is an "open source environment" section. Looks like the tasks are similar. It will be interesting to see the comparison w/ SayCan
>
> Thank you for your comments. The work SayCan aims to provide real-world grounding by means of pre-trained skills. In comparison, our work utilizes LLMs as both high-level task planners and low-level motion controllers without dependency on pre-trained skills. Therefore, it may be unfair to include SayCan as the baseline.
>
> > Is the policy able to re-try if the first trial fails?
>
> Yes. We query the LLM to re-plan sub-tasks and motion sequences every five time steps to correct any possible fails. To approve the effectiveness of re-plan, we applied our method in complex scenarios, involving potential interferences with object positions. More specifically, we randomly change the object position at any given time step within an episode. An example of environmental observation and robot trajectories is provided in Figure 6 for your reference. The results demonstrate the robustness of our method, notably its ability to dynamically recalibrate motion sequences.

---

### Official Review · Reviewer_4STn · 2023-11-01

**Soundness:** 2 fair
**Presentation:** 2 fair
**Contribution:** 1 poor
**Rating:** 3
**Confidence:** 4

**Summary:**

The authors present a training-free grounded LLM approach for Embodied AI called LLM+A(ffordance). It leverages LLM as both the sub-task planner (that generates high-level plans) and the motion controller (that generates low-level control sequences). To ground these plans and control sequences on the physical world, they develop the affordance prompting technique that stimulates the LLM to 1) predict the consequences of generated plans and 2) generate affordance values for relevant objects. Empirical evaluation is shown on robotic manipulation tasks.

**Strengths:**

1. The paper is generally well-written and easy to follow.
2. The work tackles a very challenging and useful problem for the embodied AI community -- handling high-level and low-level planning jointly with a single foundational model.

**Weaknesses:**

1. Limited technical contribution: this work comes off as an empirical evaluation of certain style of prompt engineering for robotic manipulation. Other than the fact that some prompt worked for a small set of robotic manipulation tasks, I am not sure what I learnt from this paper.
2. Limited evaluation:
 -  Furthermore, the evaluation tasks are too simple. Unclear how affordance prompting would scale to more complex or more real world tasks, for instance manipulation in cluttered settings or situations with partial observability.
 - Unclear how well LLM+A will do without a really strong llm such as GPT4. The use of GPT4 also makes it a computationally slow framework for online deployment. To that end, it would be good for the authors to report execution time/time to solve for their evaluation tasks. I encourage authors to also do real-world evaluation to really put the runtime in perspective. Lastly, I'd also like to see LLM+A performance with other opensource models like Llama2 or Vicuna.
- Also unclear if the results are reproducible given GPT4's changing capabilities over time: https://arxiv.org/pdf/2307.09009.pdf I therefore encourage authors to consider open source alternatives, at the least time-stamp their GPT4 evaluations.
3. Baselines: It is also unclear how the authors chose the baselines they compare with. They do not provide a rationale on their selection of baselines. Instead they simply choose some subset of llm prompt based approaches. Was the goal to just compare their prompt style? Why not also compare with Palm-e, RT-2, GATO, VIMA to show that their training-free prompt-based approach works better than these others that required additional data for training? Even if the goal is to compare prompt-based approaches, many others come to mind such as text2motion: https://sites.google.com/stanford.edu/text2motion
4. Limited analysis: Given that the tasks were evaluated in sim, I would have liked a more detailed failure analysis, for instance assuming perfect vision information. Authors explain that Block2Block has low success rate because of the need to reason about interaction with other blocks. But then shouldnt this be the case with SeparateBlock task as well? Also, what about Block2Position task? The success rates in Block2Position task also seem low (42%).

**Questions:**

- Why not give examples to naive llm baseline given that it needs to output coordinates in specific format too, just like LLM+A?
- Do all other baselines also use GPT4? This isnt mentioned anywhere in the paper, but I assumed this was the case. Please clarify.

---

> ### Author Response · Authors · 2023-11-22
> **Response to Reviewer 4STn (1/2)**
>
> Dear Reviewer 4STn,
>
> We greatly appreciate your insightful comments and the time you took to review our manuscript. Your feedback is essential to improving our work, and here we address your concerns:
>
> > Limited technical contribution: this work comes off as an empirical evaluation of certain style of prompt engineering for robotic manipulation. Other than the fact that some prompt worked for a small set of robotic manipulation tasks, I am not sure what I learnt from this paper.
>
> In this work, we demonstrate that a pre-trained LLM (GPT4), equipped with only an off-the-shelf detection model, can guide a robot manipulator by outputting high-level sub-task plans and low-level sequences of end-effector waypoints. This is quite a challenging task. First, the proposed method does not depend on pre-trained skills, motion primitives, or trajectory optimizers. Second, the proposed prompting method is task-agnostic so can easily generalize to heterogeneous tasks. Third, since LLMs are not trained for grounded physical interaction, the proposed affordance prompting effectively improves the executability of both the sub-task plans and the control sequences. We appreciate that you denote that the number of robotic manipulation tasks is limited in the previous manuscript. In response, we plan to conduct more realistic robotic manipulation tasks in revision, such as “open the drawer”, “push the button”, etc.
>
> > Furthermore, the evaluation tasks are too simple. Unclear how affordance prompting would scale to more complex or more real world tasks, for instance manipulation in cluttered settings or situations with partial observability.
>
> Thank you for your valuable suggestions and we agree with you that the experiment should include more complex tasks. Accordingly, we have incorporated additional testing scenarios. First, we diversified the shape of objects, introducing cubes, pentagons, star-shaped, and moon-shaped objects. We conducted an evaluation of our method over 100 episodes in the Block2Position task, resulting in a success rate of 38%. Notably, this rate is nearly consistent with the success rate reported in our paper. Besides, an example is shown in Figure 5. Second, we applied our method in complex scenarios, involving potential interferences with object positions. More specifically, we randomly change the object position at any given time step within an episode. An example of environmental observation and robot trajectories is provided in Figure 6 for your reference. The results demonstrate the robustness of our method, notably its ability to dynamically recalibrate motion sequences. Additionally, although partial observability may occur in real-world tasks, our method can address this issue through re-planning in the fixed interval.
>
> > Unclear how well LLM+A will do without a really strong llm such as GPT4. The use of GPT4 also makes it a computationally slow framework for online deployment. To that end, it would be good for the authors to report execution time/time to solve for their evaluation tasks. I encourage authors to also do real-world evaluation to really put the runtime in perspective. Lastly, I'd also like to see LLM+A performance with other opensource models like Llama2 or Vicuna.
>
> Thank you for your valuable suggestion.As the currently most powerful LLM, GPT4 encompasses plentiful commonsense and excellent abilities of reasoning and instruction following. Additionally, GPT is prone to generate outputs as we instructed that can be executed in the code. However, open-source models like Llama2 or Vicuna often throw errors both syntactically and semantically even if we show examples and instruct the output format in the prompt. Maybe instruction finetuning can alleviate this issue, but this is not the focus of our paper.
>
> To increase the time efficiency, we query GPT4 to in a fixed interval generate sub-task plans and motion sequences instead of re-planning new trajectory waypoints every time step. Specifically, the planned sequence will be updated only after the robot finishes five waypoints in the previous query round. The average query number of each episode in evaluation tasks can be referred to in the following table. We do not include the query time since it depends on different sources of OpenAI APIs.
> | Task | Block2Position | Block2Block | Separate |
> | --- | --- | --- | --- |
> | GPT4 average query number | 3.86 | 3.75 | 1.82 |

---

> > ### Author Response · Authors · 2023-11-22
> > **Response to Reviewer 4STn (2/2)**
> >
> > > Also unclear if the results are reproducible given GPT4's changing capabilities over time: https://arxiv.org/pdf/2307.09009.pdf I therefore encourage authors to consider open source alternatives, at the least time-stamp their GPT4 evaluations.
> >
> > Thank you for your valuable comments. Regarding the changing abilities of GPT4, we evaluate our method in the Block2Position task with different versions of GPT4, including March, June and August. The results are shown in the following Table. Also, we include the time-stamp of our GPT4 evaluations in the current manuscript.
> > | Task | GPT4-March | GPT4-June | GPT4-August |
> > | --- | --- | --- | --- |
> > | Block2Position | 42% | 40% | 42% |
> >
> > > It is also unclear how the authors chose the baselines they compare with. They do not provide a rationale on their selection of baselines. Instead they simply choose some subset of llm prompt based approaches. Was the goal to just compare their prompt style? Why not also compare with Palm-e, RT-2, GATO, VIMA to show that their training-free prompt-based approach works better than these others that required additional data for training? Even if the goal is to compare prompt-based approaches, many others come to mind such as text2motion: https://sites.google.com/stanford.edu/text2motion.
> >
> > We appreciate your concern regarding the selection of baselines in our study. Firstly, we would like to clarify that the choice of baselines was primarily limited by the availability of open-source code for replication. Unfortunately, many training-based methods such as Palm-e, RT-2, and GATO are proprietary and not openly accessible, which made it challenging for us to replicate and compare with these methods. Despite these limitations, we acknowledge that these training-based methods might possess an advantage in terms of final performance. However, our approach does not require training samples and demonstrates superior generalization capabilities, which makes it more accessible and easier to implement in diverse tasks. Therefore, we choose several training-free prompt-based methods for comparison. Regarding your suggestion of including text2motion as a baseline, we agree that it would be beneficial to include text2motion as a baseline for comparison. We will include the experimental results in revision..
> >
> > > Limited analysis: Given that the tasks were evaluated in sim, I would have liked a more detailed failure analysis, for instance assuming perfect vision information.
> >
> > Thank you for your comments and we append two more complex experiments to demonstrate the generalization ability of our proposed method. The additional experiments shows that our method can achieve equal performance with respect to variance in objects shapes and human interference. In our assumptions, our method doesn’t fully depend on a perfect vision information as our method plans the trajectories in a close-loop manner and it can be aware of the consequence of the action to adjust the plan. But our method can benefit from more powerful visual perception modules to help reduce the trajectory error.
> >
> > > Authors explain that Block2Block has low success rate because of the need to reason about interaction with other blocks. But then shouldnt this be the case with SeparateBlock task as well? Also, what about Block2Position task? The success rates in Block2Position task also seem low (42%).
> >
> > The Block2Block task is more difficult than the SeperateBlock from two aspects: First, the LLM should first infer the position based on the text input like “right of the blue block”.Secondly, the planned waypoints should take both two blocks into consideration to avoid collision between two blocks. For example, if the target is “put the red block to the right side of the blue block”, the interaction between the red block and the blue block can cause the red block always on the left side of the blue block. Therefore, the Block2Block task further challenges the LLM’s physical understanding of collision avoidance. In summary, the SeperateBlock requires spatial reasoning and physical understanding ability, the Block2Position further requires long-horizon planning ability and the Block2Block evaluates the collision avoidance ability additionally.
> >
> > > Why not give examples to naive llm baseline given that it needs to output coordinates in specific format too, just like LLM+A?
> >
> > Thank you for your suggestion. We have included an example of naive LLM in the Block2Postion task as shown in Listing 8. We found that Naive LLM makes many mistakes in spatial relationship reasoning. Specifically, naive LLM first fails in choosing the skills and proposes the wrong target coordinates in the subsequent step.
> >
> > >Do all other baselines also use GPT4? This isnt mentioned anywhere in the paper, but I assumed this was the case. Please clarify.
> >
> > Thank you for your comment. We confirm that all other baselines also use GPT4 and this information is clarified in the revised manuscript.

---

> > > ### Comment · Reviewer_4STn · 2023-11-22
> > > **Post rebuttal**
> > >
> > > Thanks for various explanations and additional experiments.
> > >
> > > Re: giving examples to naiveLLM. I meant that naive LLM should also be given demo1 and demo2 like the LLM+A model to ensure apples to apples comparison (including affordance values?). I wasn't asking for an example "of" naive LLM.
> > >
> > > Re: additional experiments. I appreciated additional experiments with human interference. I still find the overall set of experiments to be very simple. Also given that only GPT4 can solve these tasks, currently the main learning that I get from the paper is: GPT4 can reason about affordances and position coordinates given a structure of prompt and some examples. This is a good takeaway from a workshop paper but is not sufficient contributions for a paper IMO. So I'll keep my ratings.

---

### Official Review · Reviewer_WA1X · 2023-11-02

**Soundness:** 2 fair
**Presentation:** 2 fair
**Contribution:** 3 good
**Rating:** 5
**Confidence:** 3

**Summary:**

This works uses LLMs to generate plans that solve language-conditioned table top problems taking advantage of “affordances” prompting. The focus is very relevant for the robotics community and the benchmark is nicely selected. However, sometimes the explanation of the methods can be improved and the accuracy obtained is not in line with the claims of how affordances improve the planning.

**Strengths:**

-	Very good understanding of the challenges of LLMs solutions for robotics.
-	Introducing affordances into LLMs solution may increase generalization
-	Good selection of benchmarks.

**Weaknesses:**

-	The description of the Motion Controller could be improved as it is different in the experiments.
-	The work is relegating too much emphasis on affordance prompting, but this prompting is overengineered. For instance, Pick&place example does not shows enough the advantage of affordance prompting.
-	Results accuracy is very low comparing to other LLMs methods to solve planning.


**Focus**

I personally think that the focus is perfectly well framed in robotics and is going straight to the point to why affordances are needed. However, the example used for explaining why current methods do not work: “It may move directly right and then push the block” is not enough demonstrated with SOTA algorithms and maybe too biased by comparing with ReAct. For instance,  Driess et al. PALM-E  and interactive language can solve these type of table top problems.

**State of the art**

Previous works on affordances and tools: this could be improved. Examples:

Jamone, L., Ugur, E., Cangelosi, A., Fadiga, L., Bernardino, A., Piater, J., & Santos-Victor, J. (2016). Affordances in psychology, neuroscience, and robotics: A survey. IEEE Transactions on Cognitive and Developmental Systems, 10(1), 4-25.

Fang, K., Zhu, Y., Garg, A., Kurenkov, A., Mehta, V., Fei-Fei, L., & Savarese, S. (2020). Learning task-oriented grasping for tool manipulation from simulated self-supervision. The International Journal of Robotics Research, 39(2-3), 202-216.
Besides, what is the difference between authors approach and other LLMs table top like Interactive Language: Talking to Robots in Real Time or non-pure LLM solutions like CLIPort solution.

**Methods**

It is not totally clear for me, who is setting the object parts and how the affordance values are being generated. As this is totally different in the pushing and the pick&place experiments. Also it is not clear how the position control is generated (what is the size of the vector?, is it restricted?)

**Results**

It was not clear why the accuracy is so low despite the reasoning power of the LLM and assuming that the affordance prompting is helping out. In “language table” results are ~95% accuracy.

**Questions:**

Further comments:

Training-free means zero-shot? As the LLMs are trained.

 “Affordances” prompting is interesting but is not totally solving the problem, maybe learning non-language dynamics could be a key point for the low-level control. Otherwise LLMs, will always stay in the high-level planning.

Should be affordances as goal-conditioned values generalized non-goal-conditioned effects?

---

> ### Author Response · Authors · 2023-11-22
> **Response to Reviewer WA1X (1/2)**
>
> Dear Reviewer WA1X,
>
> We greatly appreciate your insightful comments and the time you took to review our manuscript. Your feedback is essential to improving our work, and here we address your concerns:
>
> > The description of the Motion Controller could be improved as it is different in the experiments.
>
> In the motion controller, given the decomposed sub-tasks and the affordance values from the sub-task planner, the executable motion sequences for the robots are generated. This is consistent among different tasks in our experiments. We will clarify the related descriptions to avoid possible misunderstandings.
>
> > The work is relegating too much emphasis on affordance prompting, but this prompting is overengineered. For instance, Pick&place example does not show enough the advantage of affordance prompting.
>
> We aim to explore the adaptation of the proposed affordance prompting technique. Therefore,  we do not further engineer the prompting style for each experimental task, which is verified to be effective in Pick&Place task. To further show the robustness of our method, we have incorporated additional testing scenarios. First, we diversified the shape of objects, introducing cubes, pentagons, star-shaped, and moon-shaped objects. We conducted an evaluation of our method over 100 episodes in the Block2Position task, resulting in a success rate of 38%. Notably, this rate is nearly consistent with the success rate reported in our paper. Besides, an example is shown in Figure 5. Second, we applied our method in complex scenarios, involving potential interferences with object positions. More specifically, we randomly change the object position at any given time step within an episode. An example of environmental observation and robot trajectories is provided in Figure 6 for your reference. The results demonstrate the ability of our method to dynamically recalibrate motion sequences. Additionally, we plan to conduct more realistic robotic manipulation tasks in the final version, such as “open the drawer”, “push the button”, etc.
>
> > Results accuracy is very low comparing to other LLMs methods to solve planning.
>
> Most of the previous planning-based LLM methods (such as “Language Table”) depend on either pre-trained low-level skills or massive multi-modal data to encompass diverse robotic tasks. In contrast, we utilize LLM as both the high-level sub-task planner and the low-level motion controller without the need for additional training process/data which is advantageous in many real-world applications (e.g., the scenario where the agent needs to fast adapt to heterogeneous tasks). We acknowledge that accuracy is a current shortcoming of our method, but we believe the performance of such methods will improve along with the improvement of prompt design and the base LLM.
>
> > The example used for explaining why current methods do not work: “It may move directly right and then push the block” is not enough demonstrated with SOTA algorithms and maybe too biased by comparing with ReAct. For instance, Driess et al. PALM-E and interactive language can solve these type of table top problems.
>
> We would like to explain that the proposed method can be effective in many robotic manipulation tasks except for push blocks. The affordance prompting aims to infer goal-conditioned affordance values, which indicate the executable priorities of different parts of interacted objects. For example, when instructed to “pick the pot on the table”, the robot can be guided to hold the handle of the pot with affordance prompting. We plan to report more experimental results to demonstrate this advantage.
>
> > Previous works on affordances and tools: this could be improved.
>
> Thank you for your valuable suggestions. We have included the recommended references in the current manuscript.
>
> > It is not totally clear for me, who is setting the object parts and how the affordance values are being generated. As this is totally different in the pushing and the pick&place experiments.
>
> We manually choose the object parts for different kinds of tasks. For the pushing task, we distinguish the four edges of the bounding box as object parts. For the Pick&Place task, we detect the center of the bounding box as object parts. Then, we prompt the LLMs to directly generate affordance values ranging from 0 to 1 to represent the usefulness of each object part in the sub-task planner, and an example is shown in Listing 1 in the Appendix.

---

> > ### Author Response · Authors · 2023-11-22
> > **Response to Reviewer WA1X (2/2)**
> >
> > > Training-free means zero-shot? As the LLMs are trained.
> >
> > In this paper, the zero-shot setting denotes that we do not include any example outputs in the affordance prompting for different robotic tasks. To avoid possible misunderstandings, we have revised the related descriptions in the manuscript.
> >
> > > “Affordances” prompting is interesting but is not totally solving the problem, maybe learning non-language dynamics could be a key point for the low-level control. Otherwise LLMs, will always stay in the high-level planning.
> >
> >
> > We agree with you that learning non-language dynamics is a feasible solution for low-level control. However, the dependency of massive training data and specific types of robots is a potential bottleneck. Therefore, in my opinion, it is an inspiring way to utilize the plentiful commonsense and powerful reasoning abilities of LLMs in more general robotic tasks. Since LLMs are not grounded in the physical world, affordance prompting is a potential technique for low-level motion control. Although the experimental results are still not satisfying, we will continuously improve our method to tackle failure cases in future work.
> >
> > > Should be affordances as goal-conditioned values generalized non-goal-conditioned effects?
> >
> > Thank you for your valuable suggestion. We will extend the affordance prompting to generate both non-goal-conditioned values and goal-conditioned values in our future work, which could possibly avoid the manual design of object parts.

---

### Author Response · Authors · 2023-11-22
**General Comments to All Reviewers**

We would like to thank all the reviewers for their time and effort in the review process. We’ve responded to each reviewer individually and uploaded a revised draft. In order to address concerns with the motivation and generalization concern in our paper, we have added the following experiments:
1. To verify the effectiveness of the affordance prompting, we evaluate the performance of a naive llm on the robotics control tasks, and found that it will easily make mistakes in trajectory planning because it is not grounded to the physical world
2. To simulate the diverse shape of the objects in real world,  we evaluate our method in robotics control tasks with four different shapes and found that our method can achieve equal performance compared with the main results, which demonstrate the generalization ability of our method.
3. To simulate the disturbance occurs in the real world, we evaluate the robustness of our proposed method in robotics control by introducing the interference - the target object can be moved to another position along the episode. Our LLM+A can resist to such interference by adjusting its plan in time considering the environment feedback.

---

### Meta-Review · Area_Chair_wBNN · 2023-12-10

**Metareview:**

The submission explores the use of large language models for robotic tasks, focusing on new prompting techniques such as affordance prompting.  Reviewers in general liked the idea and the writing; however, there are shared concerns about the generalization ability of the proposed method, the limited evaluation, and the weak results.  All reviewers recommended rejection.   The AC agrees and encourages the authors to revise the submission for the next venue.

**Justification For Why Not Higher Score:**

There are shared concerns about the generalization ability of the proposed method, the limited evaluation, and the weak results.

**Justification For Why Not Lower Score:**

N/A

---

### Decision · Program_Chairs · 2024-01-16

Reject